# Foreground Clustering for Joint Segmentation and Localization in Videos and Images

**Abhishek Sharma**
Navinfo Europe Research, Eindhoven, NL *
kein.iitian@gmail.com

## Abstract

This paper presents a novel framework in which video/image segmentation and localization are cast into a single optimization problem that integrates information from low level appearance cues with that of high level localization cues in a very weakly supervised manner. The proposed framework leverages two representations at different levels, exploits the spatial relationship between bounding boxes and superpixels as linear constraints and simultaneously discriminates between foreground and background at bounding box and superpixel level. Different from previous approaches that mainly rely on discriminative clustering, we incorporate a foreground model that minimizes the histogram difference of an object across all image frames. Exploiting the geometric relation between the superpixels and bounding boxes enables the transfer of segmentation cues to improve localization output and vice-versa. Inclusion of the foreground model generalizes our discriminative framework to video data where the background tends to be similar and thus, not discriminative. We demonstrate the effectiveness of our unified framework on the YouTube Object video dataset, Internet Object Discovery dataset and Pascal VOC 2007.

## 1   Introduction

Localizing and segmenting objects in an image and video is a fundamental problem in computer vision since it facilitates many high level vision tasks such as object recognition, action recognition (49), natural language description (17) to name a few. Thus, any advancements in segmentation and localization algorithm are automatically transferred to the performance of high level tasks (17). With the success of deep networks, supervised top down segmentation methods obtain impressive performance by learning on pixel level (28; 34) or bounding box labelled datasets (10; 12). Taking into account the cost of obtaining such annotations, weakly supervised methods have gathered a lot of interest lately (16; 7; 20). In this paper, we use very weak supervision to imply that labels are given only at the image or video level and aim to jointly segment and localize the foreground object given the weak supervision.

While great progress has been made in both image, video and 3D domain  (20; 5; 40) using weak supervision, most existing work are tailored for a specific task. Although UberNet (19) achieves impressive results on multiple image perception tasks by training a deep network, we are not aware of any similar universal network in the weak supervision domain that performs on both image and video data. Part of the difficulty lies in defining a loss function that can explicitly model or exploit the similarity between similar tasks using weak supervision only while simultaneously learning multiple classifiers. More specifically, we address the following challenge: how can we use semantic localization cues of bounding boxes to guide segmentation and leverage low level segmentation appearance cues at superpixel level to improve localization.

Our key idea is as follows: If an object localization classifier considers some bounding box to be a background, this, in principle, should enforce the segmentation classifier that superpixels in this bounding box are more likely to be background and vice-versa. We frame this idea of online knowledge transfer between the two classifiers as linear constraints. More precisely, our unified framework, based on discriminative clustering (2), avoids making hard decisions and instead, couples the two discriminative classifiers by linear constraints. Contrary to the conventional approach of multi-task learning (6; 24) where two (or more) similar tasks are jointly learned using a shared representation, we instead leverage two representations and enable the transfer of information implicit in these representations during a **single shot optimization** scheme.

Our work, although similar in spirit to the prior work that embeds pixels and parts in a graph (50; 29), goes a step further by modelling video data as well. To this end, we incorporate a foreground model in our discriminative clustering framework. Often, a video is shot centered around an object with similar background frames which limits the performance of discriminative clustering as shown in our experiments later. The proposed foreground model basically includes a histogram matching term in our objective function that minimizes the discrepancy between the segmented foreground across images and thereby brings a notion of similarity in a purely discriminative model. We call our method Foreground Clustering and make source code publicly available. [2]

Our contributions are as follows: 1) We propose a novel framework that simultaneously learns to localize and segment common objects in images and videos. By doing so, we provide a principled mathematical framework to group these individual problems in a unified framework. 2) We introduce a foreground model within the discriminative clustering by including a histogram matching term. 3) We show a novel mechanism to exploit spatial relation between a superpixel and a bounding box in an unsupervised way that improves the output of cosegmentation and colocalization significantly on three datasets. 4) We provide state of the art performance on the Youtube Videos Object segmentation dataset and convincing results on Pascal VOC 2007 and Internet Object Discovery Dataset.

## 2 Related work

We only describe and relate some of the existing literature briefly in this section since each of the four problems are already well explored separately on their own.

**Supervised Setting.** Numerous works (23; 47) have used off-the-shelf object detectors to guide segmentation process. Ladicky *et al.* (23) used object detections as higher order potentials in a CRF-based segmentation system by encouraging all pixels in the foreground of a detected object to share the same category label as that of the detection. Alternatively, segmentation cues have been used before to help detection (27). Hariharan *et al.* (11) train CNN to simultaneously detect and segment by classifying image regions. All these approaches require ground truth annotation either in the form of bounding boxes or segmented objects or do not exploit the similarity between the two tasks.

**Weakly Supervised Setting.** Weak supervision in image domain dates back to image cosegmentation (37; 14; 31; 18) and colocalization problem where one segments or localizes common foreground regions out of a set of images. They can be broadly classified into discriminative (14; 15; 42; 16) and similarity based approaches. Similarity based approaches (37; 44; 39; 38) seek to segment out the common foreground by learning the foreground distribution or matching it across images (38; 45; 9). All these method are designed for one of the two task. Recent work based on CNN either completely ignores these complimentary cues (20) or use them in a two stage decision process, either as pre-processing step (36) or for post processing (27). However, it is difficult to recover from errors introduced in the initial stage. This paper advocates an alternative to the prevalent trends of either ignoring these complimentary cues or placing a clear separation between segmentation and localization in the weakly supervised scenario.

**Video Segmentation.** Existing literature on unsupervised video segmentation (25; 32; 51) are mostly based on a graphical model with the exception of Brox.& Malik (4). Most notably, Papazoglou & Ferrari (32) first obtain motion saliency maps and then refine it using Markov Random Fields. Recent success in video segmentation comes mainly from semi-supervised setting (5). Semi-supervised methods are either tracking-based or rely on foreground propagation algorithms. Typically, one

initializes such methods with ground truth annotations in the first frame and thus, differ from the main goal of this paper that is to segment videos on the fly.

**Video Localization.** Video localization is a relatively new problem where the end goal is to localize the common object across videos. Prest *et al.* (35) tackles this problem by proposing candidate tubes and selecting the best one. Joulin *et al.* (16) leverages discriminative clustering and proposes an integer quadratic problem to solve video colocalization. Kwak *et al.* (22) goes a step further and simultaneously tackles object discovery as well as localization in videos. Jerripothula *et al.* (13) obtains state of the art results by first pooling different saliency maps and then, choosing the most salient tube. Most of the approaches (22; 13) leverage a large set of videos to discriminate or build a foreground model. In contrast, we segment and localize the foreground separately on each video, making our approach much more scalable.

**Discriminative Clustering for Weak Supervision.** Our work builds on the discriminative framework (2), first applied to cosegmentation in Joulin *et al.* (14) and later extended for colocalization (42; 16) and other tasks (3; 30). The success of such discriminative frameworks is strongly tied to the availability of diverse set of images where hard negative mining with enough negative(background) data separates the foreground. Our model instead explicitly models the foreground by minimizing the difference of histograms across all image frames. The idea of histogram matching originated first in image cosegmentation (37; 46). However, we are the first one to highlight its need in discriminative clustering and connection to modelling video data.

## 3 Background

In this section, we briefly review the two main components of the discriminative frameworks (14; 42; 16) used for cosegmentation and colocalization as we build on the following two components:

**Discriminative clustering.** We first consider a simple scenario where we are given some labelled data with a label vector $\boldsymbol{y} \in \{0,1\}^n$ and a $d$ dimensional feature for each sample, concatenated into a $n \times d$ feature matrix $\mathcal{X}$. We assume that the matrix X is centered. (If not, we obtain one after multiplying with usual centering matrix $\Pi = \mathcal{I}_n - \frac{1}{n}\mathbf{1}\mathbf{1}^T$). The problem of finding a linear classifier with a weight vector $\boldsymbol{\alpha}$ in $\mathbb{R}^d$ and a scalar $b$ is equivalent to:

$$\min_{\boldsymbol{\alpha} \in \mathbb{R}^d} ||\boldsymbol{y} - \mathcal{X}\boldsymbol{\alpha} - b\mathbf{1}||^2 + \beta||\boldsymbol{\alpha}||^2, \tag{1}$$

for square loss and a regularization parameter $\beta$. There exists a closed form solution for Eq1 given by: $\alpha = (\mathcal{X}^T\mathcal{X} + \beta\mathcal{I}_d)^{-1}\mathcal{X}^T\boldsymbol{y}$. However, in the weakly supervised case, the label vector $\boldsymbol{y}$ is latent and optimization needs to be performed over both labels as well as the weight vector of a classifier. This is equivalent to obtaining a labelling based on the best linearly separable classifier:

$$\min_{\boldsymbol{y} \in \{0,1\}^n, \boldsymbol{\alpha} \in \mathbb{R}^d} ||\boldsymbol{y} - \mathcal{X}\boldsymbol{\alpha} - b\mathbf{1}||^2 + \beta||\boldsymbol{\alpha}||^2, \tag{2}$$

Xu *et al.* (48) first proposed the idea of using a supervised classifier(SVM) to perform unsupervised clustering. Later, (2) shows that the problem has a closed form solution using square loss and is equivalent to

$$\min_{\boldsymbol{y} \in \{0,1\}^n} \boldsymbol{y}^T \mathcal{D} \boldsymbol{y}, \tag{3}$$

where

$$\mathcal{D} = \mathcal{I}_n - \mathcal{X}(\mathcal{X}^T\mathcal{X} + \beta\mathcal{I}_d)^{-1}\mathcal{X}^T, \tag{4}$$

Note that $\mathcal{I}_d$ is an identity matrix of dimension $d$, and $\mathcal{D}$ is positive semi-definite. This formulation also allows us to kernelize features. For more details, we refer to (2).

**Local Spatial Similarity** To enforce spatial consistency, a similarity term is combined with the discriminative term $\boldsymbol{y}^T \mathcal{D} \boldsymbol{y}$. The similarity term $\boldsymbol{y}^T \mathcal{L} \boldsymbol{y}$ is based on the idea of normalised cut (41) that encourages nearby superpixels with similar appearance to have the same label. Thus, a similarity matrix $\mathcal{W}^i$ is defined to represent local interactions between superpixels of same image. For any pair of $(a, b)$ of superpixels in image $i$ and for positions $p_a$ and color vectors $c_a$, :

$$\mathcal{W}^i_{ab} = \exp(-\lambda_p ||p_a - p_b||_2^2 - \lambda_c ||c_a - c_b||^2)$$

The $\lambda_p$ is set empirically to .001 & $\lambda_c$ to .05. Normalised laplacian matrix is given by:

$$\mathcal{L} = \mathcal{I}_N - \mathcal{Q}^{-1/2}\mathcal{W}\mathcal{Q}^{-1/2} \tag{5}$$

where $\mathcal{I}_N$ is an identity matrix of dimension $d$, $\mathcal{Q}$ is the corresponding diagonal *degree matrix*, with $Q_{ii} = \sum_{j=1}^n w_{ij}$.

## 4 Foreground Clustering

**Notation.** We use italic Roman or Greek letters (e.g., $x$ or $\gamma$) for scalars, bold italic fonts (e.g., $\boldsymbol{y} = (y_1, \ldots, y_n)^T$) for vectors, and calligraphic ones (e.g., $\mathcal{C}$) for matrices.

### 4.0.1 Foreground Model

Consider an image $I$ composed of $n$ pixels (or superpixels), and divided into two regions, foreground and background. These regions are defined by the binary vector $\boldsymbol{y}$ in $\{0,1\}^n$ such that $y_j = 1$ when (super)pixel number $j$ belongs to the foreground, and $y_j = 0$ otherwise. Let us consider the histogram of some features (e.g., colors) associated with the foreground pixels of $I$. This histogram is a discrete empirical representation of the feature distribution in the foreground region and can always be represented by a vector $\boldsymbol{h}$ in $\mathbb{N}^d$, where $d$ is the number of its bins, and $h_i$ counts the number of pixels with values in bin number $i$. The actual feature values associated with $I$ can be represented by a binary matrix $\mathcal{H}$ in $\{0,1\}^{d \times n}$ such that $H_{ij} = 1$ if the feature associated with pixel $j$ falls in bin number $i$ of the histogram, and $H_{ij} = 0$ otherwise. With this notation, the histogram associated with $I$ is written as $\boldsymbol{h} = \mathcal{H}\boldsymbol{y}$. Now consider two images $I^1$ and $I^2$, and the associated foreground indicator vector $\boldsymbol{y}^k$, histogram $\boldsymbol{h}^k$, and data matrix $\mathcal{H}^k$, so that $\boldsymbol{h}^k = \mathcal{H}^k\boldsymbol{y}^k$ ($k = 1, 2$). We can measure the discrepancy between the segmentation's of two images by the (squared) norm of the histogram difference, i.e.,

$$||\mathcal{H}^1\boldsymbol{y}^1 - \mathcal{H}^2\boldsymbol{y}^2||^2 = \boldsymbol{y}^T\mathcal{F}\boldsymbol{y}, \tag{6}$$

where $\boldsymbol{y} = (\boldsymbol{y}^1; \boldsymbol{y}^2)$ is the vector of $\{0,1\}^{2n}$ obtained by stacking the vectors $\boldsymbol{y}^1$ and $\boldsymbol{y}^2$, and $\mathcal{F} = [\mathcal{H}^1, -\mathcal{H}^2]^T[\mathcal{H}^1, -\mathcal{H}^2]$. This formulation is easily extended to multiple images (46). Since the discrepancy term in Eq. 6 is a norm, the resulting matrix $\mathcal{F}$ is positive definite by definition.

### 4.1 Optimization Problem for one Image

For the sake of simplicity and clarity, let us consider a single image, and a set of $m$ bounding boxes per image, with a binary vector $\boldsymbol{z}$ in $\{0,1\}^m$ such that $z_i = 1$ when bounding box $i$ in $\{1, \ldots, m\}$ is in the foreground and $z_i = 0$ otherwise. We oversegment the image into $n$ superpixels and define a global superpixel binary vector $\boldsymbol{y}$ in $\{0,1\}^n$ such that $y_j = 1$ when superpixel number $j$ in $\{1, \ldots, n\}$ is in the foreground and $y_j = 0$ otherwise. We also compute a normalized saliency map $M$ (with values in $[0, 1]$), and define: $\boldsymbol{s} = -log(M)$. Given these inputs and appropriate feature maps for superpixels and bounding boxes (defined later in detail), we want to recover latent variables $\boldsymbol{z}$ and $\boldsymbol{y}$ simultaneously by learning the two coupled classifiers in different feature spaces. However, to constrain the two classifiers together, we need another indexing of superpixels detailed next.

For each bounding box, we maintain a set $S_i$ of its superpixels and define the corresponding indicator vector $\boldsymbol{x}_i$ in $\{0,1\}^{|S_i|}$ such that $x_{ij} = 1$ when superpixel $j$ of bounding box $i$ is in the foreground, and $x_{ij} = 0$ otherwise. Note that for every bounding box $i$, $\boldsymbol{x}_i$ ( superpixel indexing at bounding box level) and $\boldsymbol{y}$ (indexing at image level) are related by an indicator projection matrix $\mathcal{P}_i$ of dimensions $|S_i| \times n$ such that $P_{ij}$ is 1 if superpixel $j$ is present in bounding box $i$ and 0 otherwise.

We propose to combine the objective function defined for cosegmentation and colocalization and thus, define:

$$E(\boldsymbol{y}, \boldsymbol{z}) = \boldsymbol{y}^T(\mathcal{D}_s + \kappa\mathcal{F}_s + \alpha\mathcal{L}_s)\boldsymbol{y} + \mu\boldsymbol{y}^T\boldsymbol{s_s} + \lambda(\boldsymbol{z}^T\mathcal{D}_b\boldsymbol{z} + \nu\boldsymbol{z}^T\boldsymbol{s_b}), \tag{7}$$

Given the feature matrix for superpixels and bounding box, the matrix $\mathcal{D}_s$ and $\mathcal{D}_b$ are computed by Eq. 4 whereas $\mathcal{L}_s$ is computed by Eq. 5. We define the features and value of scalars later in the implementation detail. The quadratic term $\boldsymbol{z}^T\mathcal{D}_b\boldsymbol{z}$ penalizes the selection of bounding boxes whose features are not easily linearly separable from the other boxes. Similarly, minimizing $\boldsymbol{y}^T\mathcal{D}_s\boldsymbol{y}$

encourages the most discriminative superpixels to be in the foreground. Minimizing the similarity term $\boldsymbol{y}^T \mathcal{L}_s \boldsymbol{y}$ encourages nearby similar superpixels to have same label whereas the linear terms $\boldsymbol{y}^T \boldsymbol{s_s}$ and $\boldsymbol{z}^T \boldsymbol{s_b}$ encourage selection of salient superpixels and bounding box respectively. We now impose appropriate constraints and define the optimization problem as follows:

$$\min_{\boldsymbol{y}, \boldsymbol{z}} \quad E(\boldsymbol{y}, \boldsymbol{z}) \quad \text{under the constraints:}$$

$$\gamma |S_i| z_i \leq \sum_{j \in S_i} x_{ij} \leq \eta |S_i| z_i \qquad \text{for} \quad i = 1, \ldots, m, \tag{8}$$

$$\sum_{i:j \in S_i} x_{ij} \leq \sum_{i:j \in S_i} z_i, \qquad \text{for} \quad j = 1, \ldots, n, \tag{9}$$

$$\mathcal{P}_i \, \boldsymbol{y} = \boldsymbol{x_i}, \qquad \text{for} \quad i = 1, \ldots, m. \tag{10}$$

$$\sum_{i=1}^{m} z_i = 1 \tag{11}$$

The constraint (8) guarantees that when a bounding box is in the background, so are all its superpixels, and when it is in the foreground, a proportion of at least $\gamma$ and at most ($\eta$) of its superpixels are in the foreground as well, with $0 \leq \gamma \leq 1$. We set $\gamma$ to .3 and $\eta$ to .9. The constraint (9) guarantees that a superpixel is in the foreground for only one box, the foreground box that contains it (only one of the variables $z_i$ in the summation can be equal to 1). For each bounding box $i$, the constraint (10) relates the two indexing of superpixels, $\boldsymbol{x_i}$ and $\boldsymbol{y}$, by a projection matrix $\mathcal{P}_i$ defined earlier. The constraint (11) guarantees that there is exactly one foreground box per image. We illustrate the above optimization problem by a toy example of 1 image and 2 bounding boxes in appendix at the end.

In equations (7)-(11), we obtain an integer quadratic program. Thus, we relax the boolean constraints, allowing $\boldsymbol{y}$ and $\boldsymbol{z}$ to take any value between 0 and 1. The optimization problem becomes convex since all the matrix defined in equation(7)are positive semi-definite (14) and the constraints are linear. Given the solution to the quadratic program, we obtain the bounding box by choosing $z_i$ with highest value . For superpixels, since the value of $x$ (and thus $y$) are upper bounded by $z$, we first normalize $y$ and then, round the values to 0 (background) and 1 (foreground) (See Appendix).

**Why Joint Optimization.** We briefly visit the intuition behind joint optimization. Note that the superpixel variables $\boldsymbol{x}$ and $\boldsymbol{y}$ are bounded by bounding box variable $\boldsymbol{z}$ in Eq. 8 and 9. If the discriminative localization part considers some bounding box $z_i$ to be background and sets it to close to 0, this , in principle, enforces the segmentation part that superpixels in this bounding box are more likely to be background (= 0)as defined by the right hand side of Eq. 8: $\sum_{j \in S_i} x_{ij} \leq \delta |S_i| z_i$. Similarly, the segmentation cues influence the final score of $z_i$ variable if the superpixels inside this bounding box are more likely to be foreground.

## 5 Implementation Details

We use superpixels obtained from publicly available implementation of (43). This reduces the size of the matrix $\mathcal{D}_s, \mathcal{L}_s$ and allows us to optimize at superpixel level. Using the publicly available implementation of (1), we generate 30 bounding boxes for each image. We use (26) to compute off the shelf image saliency maps. To model video data, we obtain motion saliency maps using open source implementation of (32). Final saliency map for videos is obtained by a max-pooling over the two saliency maps. We make a 3D histogram based on RGB values, with 7 bins for each color channel, to build the foreground model $\mathcal{F}$ in Eq. 6.

**Features.** Following (14), we densely extract SIFT features at every 4 pixels and kernelize them using Chi-square distance. For each bounding box, we extract 4096 dimensional feature vector using AlexNet (21) and L2 normalize it.

**Hyperparameters** Following (42), we set $\nu$, the balancing scalar for box saliency, to .001 and $\kappa, \lambda = 10$. To set $\alpha$, we follow (14) and set it $\alpha = .1$ for foreground objects with fairly uniform colors, and $= .001$ corresponding to objects with sharp color variations. Similarly, we set scalar $\mu = .01$ for salient datasets and $= .001$ otherwise.

Table 1: Video Colocalization Comparison on Youtube Objects dataset.

| Metric | LP(Sal.) | (16) | QP(Loc.) | QP(Loc.)+Seg | Ours(full) | (22) | (13)int | (13)ext |
|--------|----------|------|----------|--------------|------------|------|---------|---------|
| CorLoc. | 28 | 31 | 35 | 49 | 54 | 56 | 52 | 58 |

Table 2: Video segmentation Comparison on Youtube Objects dataset.

| Metric | LP(Sal.) | QP(Seg.) | QP(Seg. +Loc.) | Ours(full) | FST (32) |
|--------|----------|----------|----------------|------------|----------|
| IoU. | 43 | 49 | 56 | 61 | 53 |

## 6 Experimental Evaluation

The goal of this section is two fold: First, we propose several baselines that help understand the individual contribution of various cues in the optimization problem defined in section 4.1. Second, we empirically validate and show that learning the two problems jointly significantly improve the performance over learning them individually and demonstrate the effectiveness of foreground model within the discriminative framework. Given the limited space, we focus more on localization experiments because we believe that the idea of improving the localization performance on the fly using segmentation cues is quite novel compared to the opposite case. We evaluate the performance of our framework on three benchmark datasets: YouTube Object Dataset (35), Object Discovery dataset (38) and PASCAL-VOC 2007.

### 6.0.1 YouTube Object Dataset.

YouTube Object Dataset (35) consists of videos downloaded from YouTube and is divided into 10 object classes. Each object class consists of several video shots of the objects belonging to the class. Ground-truth boxes are given for a subset of the videos, and one frame is annotated per video. We sample key frames from each video with ground truth annotation uniformly with stride 10, and optimize our method only on the key frames. This is following (13; 22) because temporally adjacent frames typically have redundant information, and it is time-consuming to process all the frames. Besides localization, YouTube Object Dataset is also a benchmark dataset for unsupervised video segmentation and provides pixel level annotations for a subset of videos. We evaluate our method for segmentation on all the videos with pixel level ground truth annotation.

**Video Co-localization Experiments**

**Metric** Correct Localization (CorLoc) metric, an evaluation metric used in related work (42; 7; 22), and defined as the percentage of image frames correctly localized according to the criterion: $IoU > .5$.

**Baseline Methods** We analyze individual components of our colocalization model by removing various terms in the objective function and consider the following baselines:

**LP(Sal.)** This baseline only minimizes the saliency term for bounding boxes and picks the most salient one in each frame of video. It is important as it gives an approximate idea about how effective (motion) saliency is. We call it LP as it leads to a linear program. **Joulin *et al.*** (16) tackles colocalization alone without any segmentation spatial support. It quantifies how much we gain in colocalization performance by leveraging segmentation cues and deep features.**QP(Loc.)** only solves the objective function corresponding to localization part without any segmentation cues. So, it includes the saliency and discriminative term for boxes. **QP(Loc.)+Seg** denotes the overall performance without the foreground model and quantifies the importance of leveraging segmentation model. **Ours(full)** denotes our overall model and quantifies the utility of foreground model.

In Table 1, in addition to the baselines proposed above, we compare our method with two state of the art unsupervised approaches (13; 22). We simply cite numbers from their paper. **(13)ext** means that the author used extra videos of same class to increase the accuracy on the test video.

**Video Segmentation Experiments.** In Table 2, we report segmentation experiments on Youtube Object Dataset. We use Intersection over Union (IoU) metric, also known as Jaccard index, to measure segmentation accuracy. In addition to the stripped down version of our model, we compare

Table 3: Image Colocalization Comparison on Object Discovery dataset.

| Metric | LP(Sal.) | QP(Loc.) | TJLF14 | Ours(full) | CSP15 (7) |
|--------|----------|----------|--------|-----------|-----------|
| CorLoc. | 68 | 75 | 72 | 80 | 84 |

Table 4: Image Colocalization Comparison on Pascal VOC 2007.

| Metric | LP(Sal.) | QP(Loc.) | TJLF14 | Ours(full) | CSP15 (7) |
|--------|----------|----------|--------|-----------|-----------|
| CorLoc. | 33 | 40 | 39 | 51 | 68 |

with FST (32) which is still considered state of the art on unsupervised Youtube Object segmentation dataset.

**Discussion** We observe in both Table 1 and 2, that performance of stripped down versions when compared to the full model, validates our hypothesis of learning the two problems jointly. We observe significant boost in localization performance by including segmentation cues. Furthermore, the ablation study also underlines empirical importance of including a foreground model in the discriminative framework. On Video Colocalization task, we perform on par with the current state of the art (13) whereas we outperform FST (32) on video segmentation benchmark.

### 6.1 Image Colocalization Experiments

In addition to the baseline proposed above in video colocalization by removing various terms in the objective function, we consider the following baselines:

**Baseline Methods Tang *et al.*(TJLF14)** (42) tackles colocalization alone without any segmentation spatial support. It quantifies how much we gain in colocalization performance by leveraging segmentation cues. **CSP15** (7) is a state of the art method for image colocalization.

**The Object Discovery dataset** (38) This dataset was collected by downloading images from Internet for airplane, car and horse. It contains about 100 images for each class. We use the same CorLoc metric and report the results in Table 3.

**Pascal VOC 2007** In Table 4, we evaluate our method on the PASCAL07-6x2 subset to compare to previous methods for co-localization. This subset consists of all images from 6 classes (aeroplane, bicycle, boat, bus, horse, and motorbike) of the PASCAL VOC 2007 (8). Each of the 12 class/viewpoint combinations contains between 21 and 50 images for a total of 463 images. Compared to the Object Discovery dataset, it is significantly more challenging due to considerable clutter, occlusion, and diverse viewpoints. We see that results using stripped down versions of our model are not consistent and less reliable. This again validates our hypothesis of leveraging segmentation cues to lift the colocalization performance. Our results outperforms TJLF14 (42) on all classes. Cho *et al.*, CSP15 (7), outperforms all approaches on Pascal VOC 2007.

## 7   Conclusion & Future Work

We proposed a simple framework that jointly learns to localize and segment objects. The proposed formulation is based on two different level of visual representations and uses linear constraints as a means to transfer information implicit in these representations in an unsupervised manner. Although we demonstrate the effectiveness of our approach with foreground clustering, the key idea of transferring knowledge between tasks via spatial relation is very general. We believe this work will encourage CNN frameworks such as constrained CNN (33) to learn similar problems jointly from weak supervision and act as a strong baseline for any future work that seek to address multiple tasks using weak supervision. Optimizing the current objective function using the recently proposed large scale discriminative clustering framework (30) is left as a future work.

**Acknowledgement** Part of this work was partially supported by ERC Advanced grant VideoWorld. Many thanks to Armand Joulin for helpful discussions.

# 8 Appendix

## 8.1 Toy Example

We illustrate the spatial (geometric) constraints by a simple toy example where the image contains 5 superpixels. Global image level superpixel indexing is defined by $\boldsymbol{y} = (y_1, y_2, y_3, y_4, y_5)^T$. Also, assume that there are two bounding boxes per image and that bounding box 1, $z_1$, contains superpixel $1, 3, 4$ while bounding box 2, $z_2$, contains superpixel $1, 2, 4$. Thus, bounding box indexing for first proposal $z_1$ is defined by $\boldsymbol{x_1} = (y_1, y_3, y_4)^T$ and for $z_2$ is defined by $\boldsymbol{x_2} = (y_1, y_2, y_4)^T$. Vector $\boldsymbol{x}$ is obtained by concatenating $\boldsymbol{x_1}$ and $\boldsymbol{x_2}$. Then, vector $\boldsymbol{x_1}$ and vector $\boldsymbol{y}$ are related by $\mathcal{P}_1$ as follows:

$$\begin{bmatrix} \boldsymbol{x_1} \end{bmatrix} = \begin{bmatrix} y_1 \\ y_3 \\ y_4 \end{bmatrix} = \underbrace{\begin{bmatrix} 1 & 0 & 0 & 0 & 0 \\ 0 & 0 & 1 & 0 & 0 \\ 0 & 0 & 0 & 1 & 0 \end{bmatrix}}_{\mathcal{P}_1} \times \underbrace{\begin{bmatrix} y_1 \\ y_2 \\ y_3 \\ y_4 \\ y_5 \end{bmatrix}}_{\boldsymbol{y}}$$

Note that $|S_i| = 3$ since each bounding box contains 3 superpixels, $m = 2$ and $n = 5$.

$$\gamma |S_i| z_i \leq \sum_{j \in S_i} x_{ij} \leq (1 - \gamma)|S_i| z_i \quad \text{for} \quad i = 1$$

$$\Rightarrow \gamma * 3z_1 \leq (x_{11} + x_{12} + x_{13}) \leq (1 - \gamma) * 3z_1$$

$$\Rightarrow \gamma * 3z_1 \leq (y_1 + y_3 + y_4) \leq (1 - \gamma) * 3z_1 \text{ (By } \mathcal{P}_1 \boldsymbol{y} = \boldsymbol{x_1})$$

Similarly, the second constraint for superpixels is equivalent to:

$$\sum_{i:j \in S_i} x_{ij} \leq \sum_{i:j \in S_i} z_i, \text{for} \quad j = 1, 2, 3, 4, 5$$

$$(x_{11} + x_{21}) \leq (z_1 + z_2) \Rightarrow 2y_1 \leq (z_1 + z_2)$$

$$x_{22} \leq z_2 \Rightarrow y_2 \leq z_2$$

$$x_{12} \leq z_1 \Rightarrow y_3 \leq z_1$$

$$(x_{13} + x_{23}) \leq (z_1 + z_2) \Rightarrow 2y_4 \leq (z_1 + z_2)$$

**Rounding for segmentation** Following Wang *et al.*(45), to convert the segmentation variable into binary indicator variables, we simply sample 30 thresholds within an interval uniformly, and choose the threshold whose corresponding segmentation has the smallest normalized cut score.

## Footnotes

*work done before

[2]https://github.com/Not-IITian/Foreground-Clustering-for-Joint-segmentation-and-Localization

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
