[Reviews · NeurIPS 2018]

Reviewer 1



The authors introduce a quadratic optimization with relaxed boolean constraints that jointly learns to localize and segment objects from images and videos. So, a graph cut-based clustering framework is enhanced through transferring learning and least square loss that allows coupling foreground labels and bounding boxes. Overall, the transferring of knowledge via spatial relation is the core of the proposal and seems to be interesting to support CNN-based frameworks. The paper is clearly written, with the method detailed (both mathematically and intuitively). However, experimental results are not completely convincing, especially, those presented in table 3. Moreover, a suitable analysis regarding the free parameters in Eq (7) could be performed.

Reviewer 2



The paper is good, presents a new idea that achieves excellent results at the level of performance. I'm familiar with these types of techniques but I can not quite understand why has been used the histogram metric rather than some gradient-based technique (e.g., Gradient Edge Detection) to differentiate an object across all image. The submission present a video/image segmentation and localization framework for learns to localize and segment common objects in images and videos using a discriminative clustering model. The originality of the work is focused on: 1) Introduce a foreground/beckground discriminative model nased on histogram matching term. This foreground model are incorporated into discriminative clustering one and optimizated together. 3) Show a novel mechanism to exploit spatial relation between superpixel and bounding box in an unsupervised way. The technical content of the paper appears to be correct but the authors did not highlight well both the strengths and weaknesses of their work. It's generally well-written and structured clearly. Maybe it's too structured in the paragraph 6. In this case, I recommend making the text more fluid. I don't have any minor remarks and typos.

Reviewer 3



This paper formulate the video/image segmentation and localization problem as a single optimization problem that leverages the geometric relationship between bounding boxes (localization) and super pixels (segmentation) as optimization constraints. It also incorporate a foreground model. However, the reviewer don't see much novelty in this framework as this kind of optimization formulation has been used for this problem in prior literatures and the improvement based on modeling bbox and super pixels seems to be minor. The experimental results do not look promising either. This makes the contribution and significance of the paper to be questionable.